# Comparative Manufacturing of Hybrid Composites with Waste Graphite Fillers for UAVs

**DOI:** 10.3390/ma15196840

**Published:** 2022-10-01

**Authors:** Veena Phunpeng, Karunamit Saensuriwong, Thongchart Kerdphol

**Affiliations:** 1School of Mechanical Engineering, Institute of Engineering, Suranaree University of Technology, 111, Maha Witthayalai Rd., Suranaree, Mueang Nakhon Ratchasima District, Nakhon Ratchasima 30000, Thailand; 2Department of Electrical Engineering, Faculty of Engineering, Kasetsart University, 50 Ngamwongwan Rd., Chatuchak, Bangkok 10900, Thailand

**Keywords:** hybrid composite materials, hand lay-up, vacuum infusion, flexural strength, graphite fillers

## Abstract

Materials of Unmanned Aerial Vehicles (UAVs) parts require specific techniques and processes to provide high standard quality, sufficiently strong, and lightweight materials. Composite materials with a proper technique have been considered to improve the performance of UAVs. Usually, the hybrid composite is developed by mechanical properties with the addition of the filler component (i.e., particle) in a matrix. This research work aims to develop the effective composite materials with better mechanical properties. Considering the manufacturing of hybrid composite materials, the vacuum process is an affecting factor on mechanical properties. The comparison of the hand lay-up process (HL) and vacuum infusion process (VI) with controlled pressure and temperature are studied in this research. In addition, graphite fillers (i.e., 5 wt%, 7.5 wt%, 10 wt%, and 12.5 wt%) are added to the studied matrix. Obviously, the ply orientation is one of the factors that affects mechanical properties. Moreover, two types of ply orientation (i.e., [0°/90°]_4s_ and [−45°/45°]_4s_) are comprehensively investigated to improve mechanical properties in the three-point bending test. The experimental results show that the vacuum infusion process of ply orientation [0°/90°]_4s_ with the addition of 10 wt% graphite filler exhibits remarkable flexural strength from 404 MPa (without filler) to 529 MPa (10 wt% filler). Especially, the ply orientation of [0°/90°]_4s_ has higher flexural strength than [−45°/45°]_4s_ in both processes. Considering the failure, the fracture of the specimen propagates along the trajectory of fiber fabric orientation, leading to the breakage. Subsequently, the flexural strength under the vacuum infusion process is more significant than in the hand lay-up process. Effectively, it is found that the hybrid composite in this manufacturing has a higher strength-to-weight ratio to use in the structure of UAV instead of pure aluminum. It should be noted that the proposed hybrid composite strategy used in this study is not only limited to the UAV parts. The contribution can be extended to use in other applications such as automotive, structural building, and so on.

## 1. Introduction

Composite materials combine two or more different structures or chemical compositions to improve their properties [1,2]. Generally, composite materials consist of a matrix and fiber, which serve as the content and reinforcement, respectively [3,4,5]. When two or more types of materials are combined (e.g., fiber and metal, fiber and ceramic, fiber and polymer or metal and metal), several effects need to be observed using experimental or numerical analysis [6,7]. Nowadays, composite materials are developed on the model of structural changes in elements (i.e., fiber, matrix, and particle) to improve the properties of composite materials. Therefore, composite materials are used as materials for ships, automobiles, aircrafts, and UAVs [8,9]. Up to this date, the composites are used as components of the UAV to improve the performance of the UAV in achieving various missions controlled by a mathematical model program (i.e., fuzzy control system and PID system) [10]. Furthermore, composites are selected for using in UAVs because of their mechanical properties. When the material has high mechanical properties and is lightweight, it results in a higher strength-to-mass ratio, indicating the strength of composite materials [11].

Mechanical properties are very significant. Before determining an appropriate material for work, the initial process to consider is the mechanical properties. To examine the mechanical properties, various methods of testing (i.e., tensile tests, bend tests, and compression tests) are performed in the engineering aspect [12]. Focusing on the aviation industry, the determination of the studied mechanical properties depends on the interesting values and the application of the material, in which aerospace requires high flexural strength [13]. The bending of the wing can be affected by the lift force and fixed components against with fuselage. For this reason, the materials that are applied to the wings should have a high flexural strength to be subjected to their expected loads. Therefore, the bending tests are used in accordance with the American Society for Testing and Materials (ASTM) with respect to mechanical properties. However, composite materials have been deliberately developed to increase mechanical properties [14,15]. These mechanical properties can be improved by the addition of fiber or matrix of a composite material [15,16]. For fibers, the various fibers (i.e., carbon fiber, aramid fiber, and glass fiber) in composite materials can greatly influence such mechanical properties [17]. By changing the types of matrix (i.e., epoxy resin, metal, and polymer) or adding two or more types of fibers, these modifications can affect the mechanical properties. In fact, the different types of fibers have dissimilar diameters, which can lead to a significant gap between the fiber and matrix, resulting in some limitations and weaknesses of hybrid composite. To reduce the air gap, sufficient fillers (i.e., graphite filler and micro balloon) can be added in the matrix, enhancing the higher mechanical properties [18,19,20,21]. Megahed et al. [22] investigated the addition of a graphite filler to carbon fiber/epoxy composites and glass fiber/epoxy composites, which are referred as hybrid composites. When a graphite filler is added, the mechanical properties are increased compared to those without graphite fillers. By adding graphite fillers to both composites, it was found that the addition of a graphite filler to carbon fiber/epoxy composites increased mechanical properties more than the addition of glass fiber/epoxy composites. Thus, adding a graphite filler to carbon fiber/epoxy composites could improve the fiber and matrix coordination [23,24,25]. Krishnaraj et al. [26] studied hybrid composite characteristics based on two types of toner (i.e., waste toner and carbon black nanopowder). Both toners were heated to improve the bonding capability with other materials. The waste toner is also one of the fillers to be used in composite materials. Subramani et al. [27] investigated the waste toner in the part of carbon/Fe_3_O_4_ nanocomposites for producing capacitors. The capacitor produced by the waste toner filler could substantially increase the efficiency of electrical conductivity with excellent electrochemical stability.

Carbon fiber is one of the fiber types that are very popular due to its lightweight and high strength compared to other types of fibers [28,29]. Due to the aforementioned properties, the carbon fiber is used in the aviation industry, automotive industry, and marine industry [30,31]. Considering the crack propagation of the carbon fiber/epoxy composites, the direction of the fiber has to be designed for the sufficient loads corresponding to the subjected force. The fracture of carbon fiber/epoxy composites can be caused by the fabric pattern [32,33,34,35], so the ply orientation is a substantial factor that affects the mechanical properties. Richardo et al. [36] used the carbon fiber fabric in directions [0°/90°]_4s_ and [−45°/45°]_4s_ with graphite filler at 0, 7.5, 10, and 11.5 wt% to form carbon fiber/epoxy composites for testing the mechanical properties. At such ply orientation [0°/90°]_4s_, the flexural strength was higher than [−45°/45°]_4s_, and the shear test at ply orientation [−45°/45°]_4s_ was able to withstand shear strength better than the ply orientation [0°/90°]_4s_. Another significant factor affecting mechanical properties depends on the manufacturing of composite materials as the manufacturing process has to be controlled by a sufficient environment (i.e., pressure and temperature). The fracture of hybrid composites over 11.5 wt% indicates a macropore and increases porosity [37].

Considering the manufacturing, it is a considerable factor affecting the mechanical properties of composite materials [38,39,40]. Depending on the application, there are several methods for manufacturing composites (i.e., hand lay-up process, vacuum infusion process, and compression process). In aviation applications, the vacuum infusion process is used to reduce air bubbles in the materials, resulting in high strength [41,42,43]. Kim et al. [44] studied the manufacturing process of glass fiber/epoxy composites between the hand lay-up with vacuum bagging and vacuum infusion. Evidentially, the hand lay-up process represents simplicity, leading to the reduction in resources. It is widely used in the production of composite materials. Nevertheless, the controlled environmental conditions of the manufacturing (i.e., curing vacuum pressure and temperature profile) directly affected the mechanical properties of composite materials. As evidence, the curing temperature at 100 °C for 2 h was applied to improve the mechanical properties of the carbon fiber/epoxy composite. Hoda et al. [45] studied the carbon fiber/epoxy composites via the control pressure and temperature to determine the optimal pressure and temperature, leading to superior mechanical properties. When the vacuum pressure is decreased, the mechanical properties improve as the air gap is reduced in the materials with the optimum curing temperature at 100 °C. This evidence indicates the environment could affect the mechanical properties [46,47,48].

Therefore, this research aims to enhance the mechanical properties of the hybrid composites laminates from wastes of toner cartridges (i.e., carbon fiber/epoxy composites with graphite fillers) and to compare the manufacturing of the vacuum bagging process and vacuum infusion process with controlled pressure and temperature. To investigate the sensitivity of the composites for higher flexural properties, this study validates the alterations of the ply orientation and angle of the carbon fiber fabric. Finally, the hybrid composite material with adding the graphite filler from waste in this study can contribute to the further manufacturing of the preliminary UAV structure (i.e., drone cover, rib, spar, and frame) instead of using aluminum.

## 2. Materials and Methods

### 2.1. Materials

In this section, the 3K carbon fiber fabric made by Toray Carbon Fibers American, Inc. (see Figure 1a), with the 1 × 1 plain weave corresponding to the weight of 200 g/m^2^ and density of 1.8 g/cm^3^, is used to prepare for manufacturing the studied hybrid composites. The matrix is made of the epoxy resin ER550 with the resin and hardener ratio of 100:35, respectively. The graphite filler with the average size of 5 μm diameter from wastes of a printer toner cartridge is measured by the Scanning Electron Microscope (SEM) analysis (see Figure 1b). The vacuum processes are compared between vacuum bagging and vacuum infusion to determine the difference in mechanical properties. For decades, the vacuum bagging process has been introduced and widely used in several industries, especially in UAV components. Due to its ability, the vacuum bagging process yields a more consistent, thinner, lighter, and stronger product than aluminum and other metals. However, up to this date, the vacuum infusion process has been widely used in the cutting-edge technology for many industries. Outstandingly, the vacuum infusion process brings all of the environmental advantages of a closed mold process. The resin curing in a closed environment leads to an excellent fiber-to-resin ratio with minimal-to-no voids in the finished laminate. In this paper the vacuum process is not only compared, but also the effect of waste graphite filler is introduced.

### 2.2. Prepared Specimens

Considering the preparation of the carbon fiber fabric, the angles of the fabric are cut into the specific sizes of 25 × 25 cm^2^ at −45°, 45°, 0°, and 90° for 40 sheets per direction. The structure of the fabric is displayed in Figure 2. To mix the filler with the epoxy, this study first combines the epoxy resin with the graphite filler at a ratio of 5, 7.5, 10, and 12.5 wt%. Then, the mixed epoxy resin is gently stirred for a few minutes. After a certain time of stirring, the hardener is poured into the mixture and gently stirred in one direction to avoid the air bubble. To conduct the studied hybrid composite materials, the specimen designs are described in Table 1.

In addition to the basic selection of the filler percentage based on [34], the behavior of the improved hybrid composite can be accepted around 10–12.5 wt% of graphite filler. Thus, to observe the wide range of hybrid composite behavior, this study selected 5, 7.5, 10, and 12.5 wt% of the filler content. In addition to the homogenous and agglomeration in the mixing process, it could not be confirmed that the mixing would provide perfectly homogeneous and agglomerate. However, in the experiment this study controlled the significant agglomeration factors (i.e., mixing ratio, mixing time, and stirring rate) at the same condition. This could improve the mechanical properties because the filler provides a bridge between fiber and matrix. In addition to the environment of the curing process, this research considers two different processes: (1) hand lay-up with vacuum bagging; and (2) vacuum infusion. Both processes are controlled by the pressure at −0.8 bar and temperature at 100 °C. However, in this study the humidity is not controlled because the composite did not expose to the air. In fact, the composite is still in the vacuum bag film, which can be used at temperatures up to 170 °C all along the curing time in an oven.

### 2.3. Hand Lay-Up Process

In this process, the manufacturing relies on the hand lay-up process. The aluminum plate is used as a mold for manufacturing the carbon fiber/epoxy laminate layer, where eight layers of carbon fiber fabric with the epoxy resin mixture are laid as described in Table 1. After that, the peel ply, release film, and breather are laid to the carbon fiber/epoxy laminate, respectively. Then, the manufacturing of the laminate has been completed. The laminate is placed in the vacuum bagging and removed the air by the vacuum pump, as shown in Figure 3. To control the environment of the manufacturing, the pressure is assigned at −0.8 bar, and the hybrid composite laminates are cured using the OV301 curing oven at 100 °C with a temperature profile as shown in Figure 4.

### 2.4. Vacuum Infusion Process

Vacuum infusion molding uses a glass plate mold to manufacture carbon fiber/epoxy laminates. In this process, eight layers of the carbon fiber fabric are placed, followed by the peel ply, infusion net, and vacuum bagging sealed by a sealant tape. The temperature profile is shown in Figure 4. The epoxy resin with/without graphite is added to the laminates under the vacuum pressure, as shown in Figure 5. The vacuum pressure was controlled at −0.8 bar for 15 min and dried for two days. Then, the hybrid composites are dried and cured by the curing oven (i.e., OV301) at 100 °C.

### 2.5. Testing Specimens

Testing for mechanical properties can be performed by a variety of testing methods. In UAVs, it is obvious that the wing and other structures are practically subjected to a complex combination of forces, including tension, compression, and shear. In addition to aerospace applications, there are many forces acting on structures with significant force; that is, the bending force caused by the lift and friction of the structures. Thus, this study performs the flexural test to understand aspects of the material’s behavior under the flexural load. Therefore, this research pays special attention to the bending test to determine flexural strength. This study used the three-point bending test (see Figure 6) with a Universal Testing Machine (UTM) of 100 kN. Testing is performed in accordance with ASTM D790-02 [49,50] with a crosshead speed of testing of 5 mm/min and a span length of 100 mm. The test specimen measured 191 × 20 × 2 mm^3^ (see Figure 7). The flexural strength can be calculated by the following equation:(1)$$\sigma =3FL/2bt^{2}$$
where *σ* is the flexural strength, *F* is the maximum load subjected to the specimens of the test, *L* is the span length between the supports, *b* is the width of a specimen for testing, and *t* is the thickness of a specimen for testing.

## 3. Experimental Results and Discussion

### 3.1. Hand Lay-Up Process

The hand lay-up manufacturing is the process to create the laminate, where the air gap is sufficiently reduced, thus improving the strong bonding between the matrices. Subsequently, the maximum load is increased by the coordination between the matrix and reinforcement. In this study, the maximum load applied to the specimen can be obtained by the three-point bending test. Figure 8 shows the maximum load of specimens using the hand lay-up process. As a result, in the orientation of [0°/90°]_4s_ the maximum load applied to the hybrid composites is increased by the addition of the graphite filler. The maximum load at the ply orientation of [0°/90°]_4s_ is increased by 15.7%, 48%, and 55.8% with the addition of the graphite filler at 5, 7.5, and 10 wt%, respectively, compared to the carbon fiber/epoxy without graphite (0 wt%). Evidently, the tendency graph of maximum load at the ply orientation of [0°/90°]_4s_ is similar to the tendency graph of ply orientation [−45°/45°]_4s_. In the ply orientation of [−45°/45°]_4s_, it is obvious that the percentages of the graphite at 5, 7.5, and 10 wt% affect the maximum load by 2.4%, 14.9%, and 28.7%, compared to the carbon fiber/epoxy without graphite (0 wt%). Considering the graphite with polymer, the curing heats the graphite to increase the bonding capability between the fiber and matrix. This process can significantly improve the adhesion between the fiber and matrix, resulting in higher mechanical properties. Compared with 12.5wt%, the additional graphite is considered at the ply orientations of [0°/90°]_4s_ and [−45°/45°]_4s_. Consequently, there is a substantial decrease in the maximum load structure applied to the specimen by 10.9% and 9.6%, respectively. Obviously, the maximum load decreases as the addition of the increasing graphite filler could affect the agglomerates, causing porosity in the matrix. Therefore, it is found that the adhesion between the matrix and reinforcement represents the weakness to the hybrid composite.

The relationship between the flexural strength and percentage of graphite is analyzed in Figure 8. As a result, when the load is applied to the specimen via the three-point bending test, the flexural strength can be determined. By analyzing the flexural strength, it can be observed that the ply orientation affects the mechanical properties at the ply orientations of [0°/90°]_4s_, which possess higher flexural strength rather than the orientation of [−45°/45°]_4s_. By placing the orientation of [0°/90°]_4s_ with/without the graphite filler at 0, 5, 7.5, 10, and 12.5 wt%, the results show the augmentation of flexural strength by 138.5%, 169.5%, 207.3%, and 188.7%, respectively, compared to the ply orientation of [−45°/45°]_4s_. This is because the ply orientation of [0°/90°]_4s_ has a plain weave between vertical and horizontal. Subsequently, when the force is applied by bending test, the perpendicular force inside such ply orientation can absorb the flexural strength. It can be seen that the angle of [−45°/45°]_4s_ represents the oblique angle of the fabric when the shear is applied to specimens, resulting in the resistance in the shear force.

### 3.2. Vacuum Infusion Process

Figure 9 shows the maximum load affected on the specimen via the vacuum infusion process. As a result, when the percentage of graphite fillers is changed to 5, 7.5, 10, and 12.5 wt%, the maximum load increases, similar to the hand lay-up process. In the case of the ply orientation of [0°/90°]_4s_, when graphite is added by 5, 7.5, 10, and 12.5 wt%, the maximum load can be increased by 13%, 33%, 72.2%, and 28% compared to the case of the carbon fiber/epoxy without graphite (0 wt%). In the case of the ply orientation of [−45°/45°]_4s_, the increased graphite percentage of 5, 7.5, 10, and 12.5 wt% can affect the maximum load, corresponding to the augment of 14.3%, 36.3%, 41%, and 28% compared to the case of the carbon fiber/epoxy without graphite (0 wt%). Compared to the case of the additional graphite with 10 wt%, the case of 12.5 wt% of the ply orientation of [0°/90°]_4s_ and [−45°/45°]_4s_ yield a significant decrease in the maximum load applied on the specimen by 25.7% and 17.9%, respectively.

The relationship between the flexural strength and percentage of graphite fillers is shown in Figure 9. When the ply orientations are compared, the orientation of [0°/90°]_4s_ yields higher flexural strength than the orientation of [−45°/45°]_4s_. It is signified that the ply orientation can directly affect the mechanical properties. The flexural strength is increased by combining the graphite fillers at ply orientations of [0°/90°]_4s_, while the graphite fillers are added by 5, 7.5, and 10 wt%. Subsequently, the flexural strength is increased by 6%, 22%, and 31%, compared with 0 wt% of the graphite. By adding 12.5 wt% of graphite, the flexural strength is significantly reduced by 25.8% compared to the 10 wt% of the graphite. The tendency of flexural strength on the ply orientation of [−45°/45°]_4s_ is similar to the ply orientation of [0°/90°]_4s_ with graphite fillers of 5, 7.5, and 10 wt%. Consequently, the flexural strength increased by 10.6%, 20.4%, and 35.6%, when compared to the case without graphite filler. When the graphite filler is added greater than 10 wt% (i.e., 12.5 wt%), the flexural strength can be minimized by comparing the flexural strength at 10 wt% percentage of the graphite filler. Moreover, in the case of ply orientation [−45°/45°]_4s_ with 10 wt% and 12.5 wt%, the flexural strength is decreased by 23.7%. As a result, it is signified that increasing the amount of graphite fillers did not always augment the flexural strength. In fact, the highest strength point relies on the optimal content of different fillers. In this case, the optimal content of the studied filler could be found at around 10 wt% graphite filler.

The relationship between the manufacturing processes and flexural strength shows that the hand lay-up process has less strength than the vacuum infusion process in both aspects of the ply orientation and percentage of the graphite filler, as shown in Figure 10. Considering the ply orientation of [0°/90°]_4s_ at 0, 5, 7.5, 10 and 12.5 wt% of the graphite fillers based on the vacuum infusion process, it is obvious that the flexural strength can be increased by 43.1%, 31.2%, 18.3%, 20.3%, and 0.2% compared with the hand lay-up process. Together with the ply orientation of [−45°/45°]_4s_, the flexural strength based on the vacuum infusion process is increased by 20.8%, 30.6%, 26.7%, 27.4%, and 7.5% with the additional graphite filler of 0, 5, 7.5, 10 and 12.5 wt%, respectively, compared with the hand lay-up process.

### 3.3. Comparison with Aluminum

Generally, most of the materials used in the manufacturing process for the UAV constructions are made of aluminum, which is heavy and strong. In this research, the flexural strength of aluminum is comprehensively compared with the hybrid composite materials. Table 2 shows the flexural strength between the pure aluminum alloy 6061 and hybrid composite materials (i.e., 10 wt% of [0°/90°]_4s_). According to Table 2, the hybrid composite materials demonstrate higher flexural strength in the manufacturing process compared to the pure aluminum alloy 6061. However, the strength is considered in conjunction with the design weight. The weight of the specimen material is taken to determine the strength-to-weight ratio. As a result, hybrid composites produce higher strength-to-weight ratios than aluminum. By considering the comparison of the flexural strength regarding the HL and VI manufacturing, it can be seen that the VI process provides the higher flexural strength than the HL process. This is because the VI process represents a pressurized vacuum process in removing the entire air in the laminate and replacing with the epoxy resin or filler. Thus, it is confirmed that hybrid composite materials can be used as an alternative substance for producing that can be used as structural UAVs.

### 3.4. Fracture Characteristics

Concerning the strength of the material, the fracture of the material is also clarified in this subsection. This analysis can reveal the fractures of hybrid composite materials. The crack caused by force on the specimens is performed by the three-point bending test. By applying the testing force, the crack patterns on specimens are discovered, as shown in Figure 11. The studied specimen is tested by the manufacturing process with coordination between the reinforcement (i.e., carbon fiber fabric) and matrix (i.e., epoxy resin). Considering the failure, the fracture of the specimen cracks following the trajectory of fiber orientation. In the composite laminates, macro–micro images are provided in Figure 12 to observe the crack and failure area.

As illustrated in Figure 13, the carbon fiber fractures are investigated by the scanning electron microscope (SEM) with the commercial model of JOEL JSM-6010LV. At 10 wt% of the graphite filler (see Figure 13a,b) it can be seen that the matrix formulates as a group, indicating the ductility of the matrix; whereas, the matrix at 12.5 wt% of the graphite filler (see Figure 13c,d) disseminates in small pieces corresponding to the flexural strength or less ductility (see in Figure 10), signifying the degradation of strength. In the case of the vacuum infusion process (see Figure 13a,c), it can be seen that the matrix formulates as a group, indicating the strong bonding between the fiber and matrix when subjected to loads. In addition, it is obvious that the fibers are held together with less gap, while the vacuum bagging has fewer matrix components at the fracture (see Figure 13b,d). Therefore, the vacuum infusion process at 10 wt% of the graphite filler yields better performance in flexural strength than the vacuum bagging process.

It should be noted that the energy dispersive X-ray spectroscopy (EDS) represents several physical phenomena corresponding to the electron interactions used for imaging sample surfaces. It is possible to take advantage of these interactions to obtain chemical information, where the EDS detector is a tool for measuring the energy of the emitted photons in the X-ray electromagnetic spectrum. However, this study is not relevant to the investigation of the chemical information or energy amount. In fact, this work pays attention to the hybrid structural analysis with flexural strength. Thus, the EDS analysis is not considered in this study.

## 4. Conclusions

In this study, the hybrid composite materials are introduced by the combination of carbon fiber/epoxy with waste graphite fillers. The main objective is to expand/promote the utilization of hybrid composite materials (e.g., waste graphite fillers) into the UAV application, instead of using the conventional material, i.e., aluminum. Thus, the mechanical properties of hybrid composite materials are comprehensively analyzed based on the three-point bending tests. As a result, it is found that the hybrid composite materials provide higher flexural strength and strength-to-weight ratio than the pure aluminum alloy 6061. Obviously, the additional waste graphite fillers exhibit a significant factor, improving the flexural strength. At 10 wt% of the filler, it reveals the highest flexural strength in the case of ply orientation of [0°/90°]_4s_ and of [−45°/45°]_4s_. Visibly, the waste graphite fillers at 10 wt% compromising in fiber composites with the ply orientation of [0°/90°]_4s_ provide remarkable properties. Therefore, the study’s waste toner (graphite filler) could improve the mechanical properties as the graphite filler successfully reduces the air gap in the lamination. In addition, employing the contaminated graphite fillers from waste toner cartridges can effectively reduce and reuse the large abundance of waste and practically eliminate the waste from commercial and industrial systems. This utilization could also diminish hazardous waste and pollution in both local and global environments. Moreover, this study shows that the ply orientation symbolizes the important factor affecting the flexural strength and load. The ply orientation directly alters the fracture characteristics of specimens from the testing. Nevertheless, the manufacturing process is also one of the factors impacting the flexural strength. The performed experiment indicates the specimens triggered by the vacuum infusion process embody the higher flexural strength. Subsequently, the hybrid composite material has a higher strength-to-weight ratio than the pure aluminum alloy, leading to the incredibly strong and lightweight materials. This study also confirmed that the hybrid composite material from the waste toner could be used to manufacture various parts of industries, including automobile and aviation today and in future, resulting in the structure’s strength enhancement with weight reduction. In future works, this application can be used as an alternative material for manufacturing UAV parts, such as ribs, wings, and fuselage. Moreover, different manufacturing processes to conduct specific parts of the UAV body should be analyzed as further work of this study.

## Figures and Tables

**Figure 1 materials-15-06840-f001:**
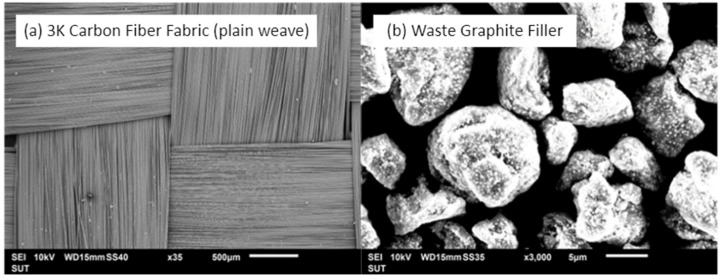
SEM photograph of the expanded view: (**a**) 3K carbon fiber fabric-plain weave, and (**b**) waste graphite filler.

**Figure 2 materials-15-06840-f002:**
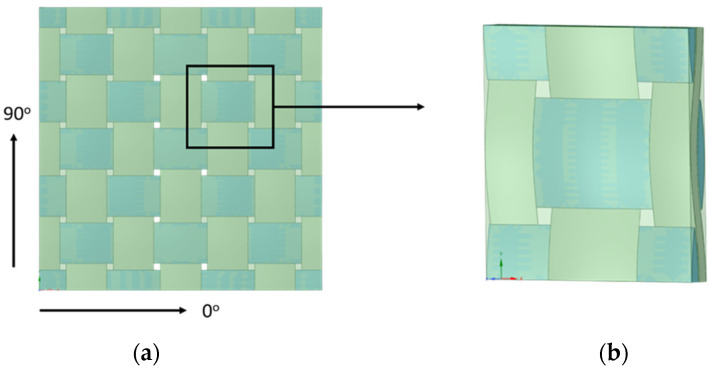
Structure of the carbon fiber fabric with: (**a**) direction 0° and 90°; (**b**) zoom-in view.

**Figure 3 materials-15-06840-f003:**
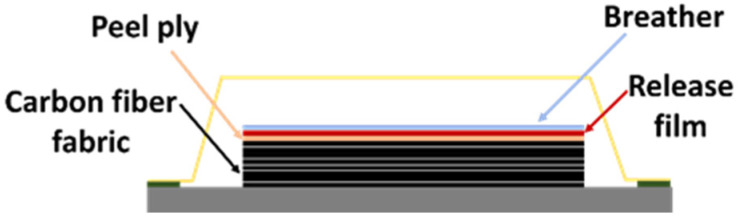
The studied laminate with the aluminum mold.

**Figure 4 materials-15-06840-f004:**
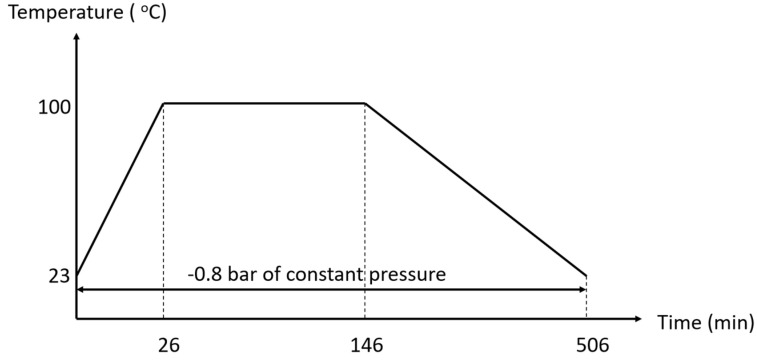
The temperature profile for curing hybrid composites.

**Figure 5 materials-15-06840-f005:**
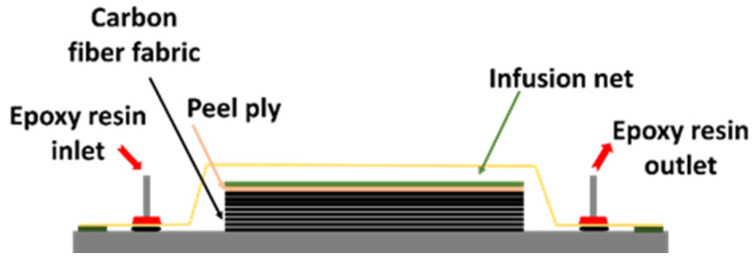
The studied vacuum infusion process.

**Figure 6 materials-15-06840-f006:**
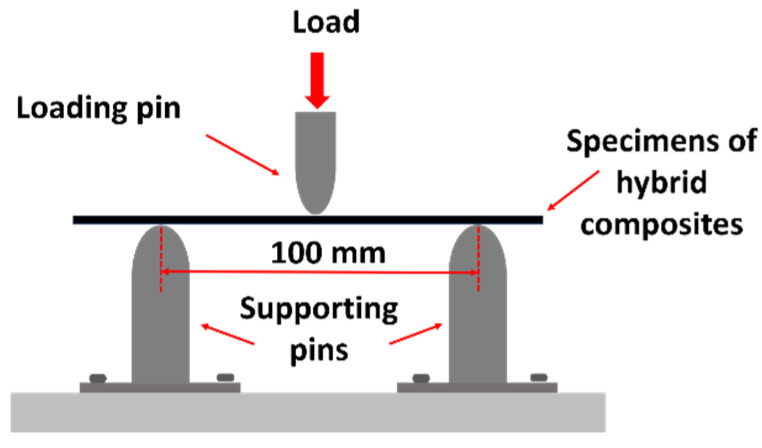
Three-point bending testing.

**Figure 7 materials-15-06840-f007:**
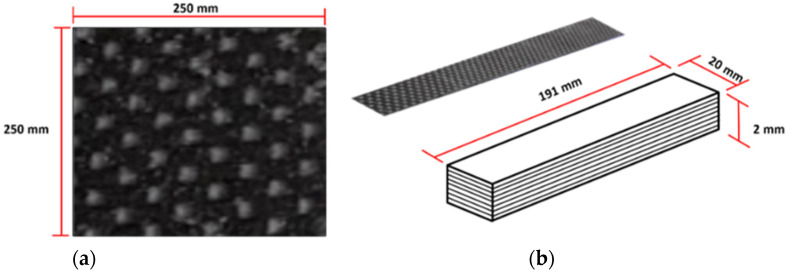
Hybrid composite laminate: (**a**) specimen before cutting; (**b**) cut specimens for the three-point bending test.

**Figure 8 materials-15-06840-f008:**
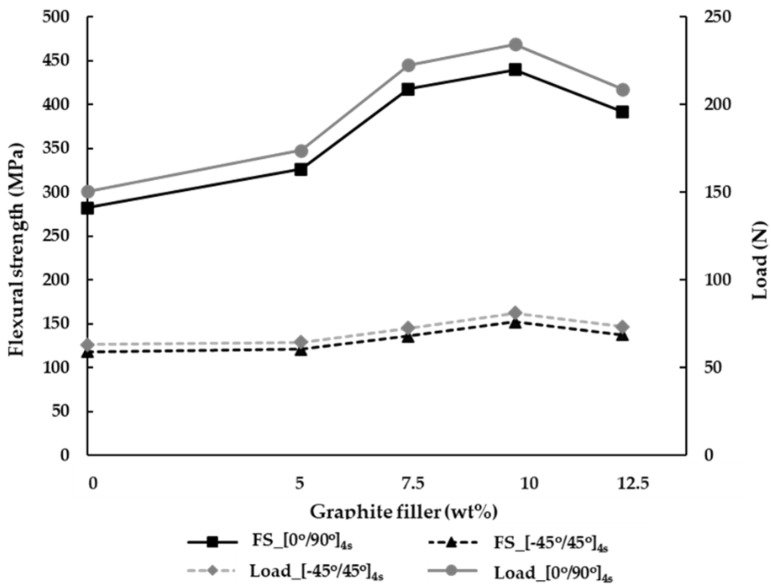
The flexural strength (FS) and maximum load of specimens using the hand lay-up process.

**Figure 9 materials-15-06840-f009:**
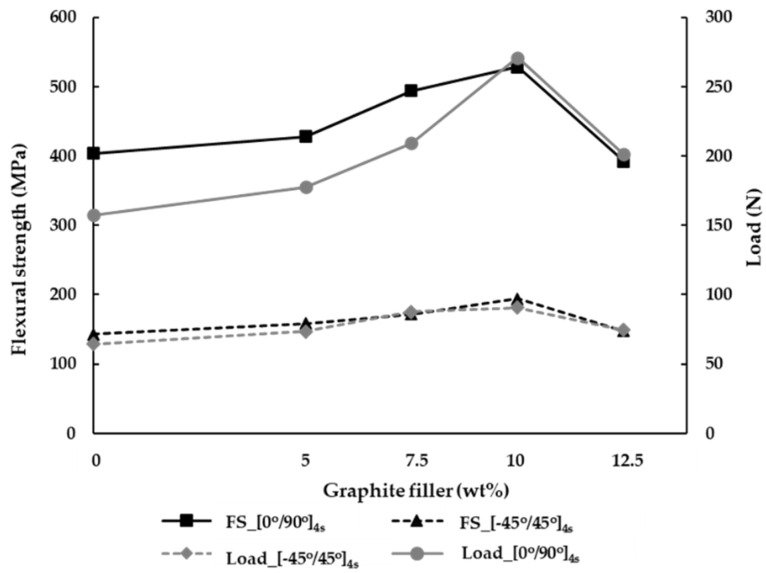
The flexural strength (FS) and maximum load affected on the specimen using the vacuum infusion process.

**Figure 10 materials-15-06840-f010:**
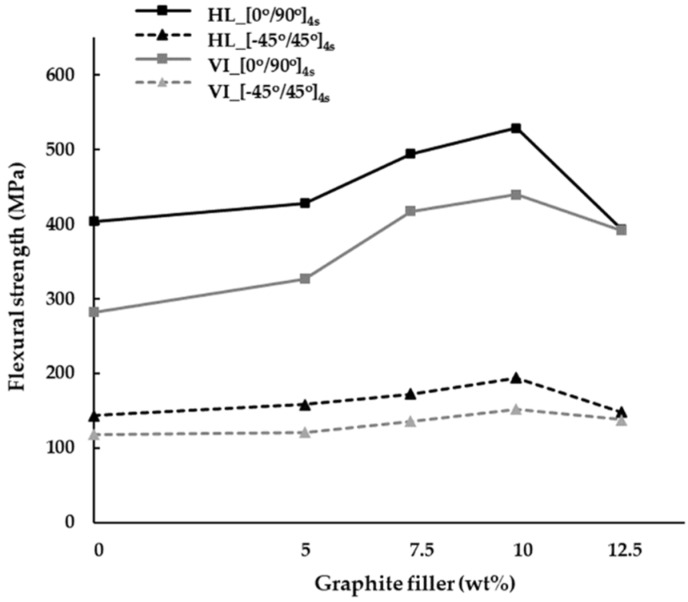
The flexural strength of different manufacturing processes.

**Figure 11 materials-15-06840-f011:**
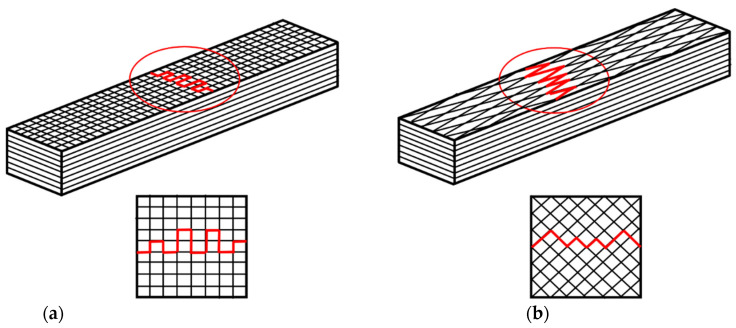
The fracture characteristics of the studied hybrid composites: (**a**) [0°/90°]_4s_; (**b**) [−45°/45°]_4s_.

**Figure 12 materials-15-06840-f012:**
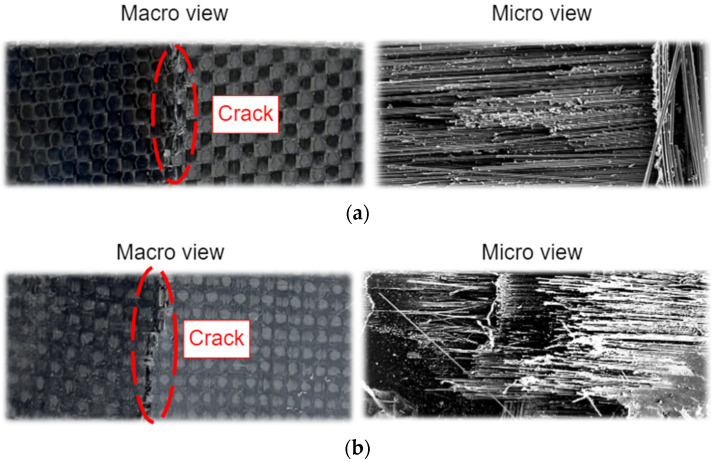
Macro-micro images of the samples with failure: (**a**) [0°/90°]_4s_; (**b**) [−45°/45°]_4s_.

**Figure 13 materials-15-06840-f013:**
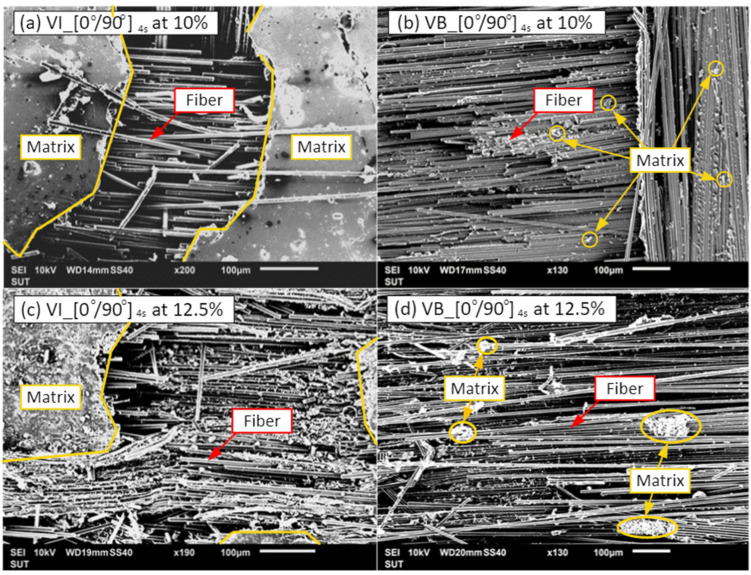
SEM of fracture specimens of [0°/90°]_4s_ with: (**a**) 10 wt% of the graphite filler under a vacuum infusion process; (**b**) 10 wt% of the graphite filler under a vacuum bagging process; (**c**) 12.5 wt% of the graphite filler under a vacuum infusion process; and (**d**) 12.5 wt% of the graphite filler under a vacuum bagging process.

**Table 1 materials-15-06840-t001:** Experimental designs of the studied hybrid composites.

Specimens	Reinforcement	Graphite Filler (wt%)	Process
1	[0°/90°]_4s_	0	Hand lay-up (HL)
2		5	
3		7.5	
4		10	
5		12.5	
6	[−45°/45°]_4s_	0	
7		5	
8		7.5	
9		10	
10		12.5	
11	[0°/90°]_4s_	0	Vacuum infusion (VI)
12		5	
13		7.5	
14		10	
15		12.5	
16	[−45°/45°]_4s_	0	
17		5	
18		7.5	
19		10	
20		12.5	

**Table 2 materials-15-06840-t002:** Strength to mass ratio of materials.

Materials	Flexural Strength (MPa)	Weight (g)	Strength-to-Mass Ratio (MPa/g)
Pure aluminum Alloy 6061	187.5	27	6.9
Hybrid composites by HL	439.4	11.75	37.4
Hybrid composites by VI	528.6	10	52.9

## Data Availability

Data sharing is not applicable to this article.

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
