# Peer review of "Comparative Manufacturing of Hybrid Composites with Waste Graphite Fillers for UAVs"

_materials, 2022, doi:10.3390/ma15196840_

Round 1
Reviewer 1 Report
Author investigated comparative Manufacturing of Hybrid Composites with Waste Graphite fillers for UAVs. Work is interesting and useful to composites application in industries but scientific depth of knowledge is missing.
1. There are some limitations in the field of hybrid composites; author should also discuss the same in the paper.
2. In Conclusion, State your findings in quantitative way and suggestions for future work.
3. What is the basis of selection of filler percentage
4. Why only bending test was selected. What about tensile, impact and other tests.
5. SEM analysis of fracture specimens after bending test.
6. Describe effect of process on bonding using surface morphology.
7. Explain with EDS analysis
8. The following latest research papers related to mechanical properties should be added in the manuscript.
a) Wankhade, Lalit N., Digvijay Rathod, Masnaji R. Nukulwar, Eshan S. Agrawal, and Ganesh R. Chavhan. "Characterization of aluminium-silicon carbide metal matrix composites." Materials Today: Proceedings (2021), Volume 44, Part 1, 2021, Pages 2740-2747, https://doi.org/10.1016/j.matpr.2020.12.699
b) E. Agrawal, V. Tungikar, Study on tribological properties of Al-TiC composites by Taguchi method, Mater. Today Proc. 26 (2020) 2242–2247. https: doi:10.1016/j.matpr.2020.02.486.
c) Sudhir W. Burande, Deepak V. Bhope,“ Evaluation and validation of mechanical properties of steel strip embedded unidirectional E-glass fibre reinforced hybrid laminate," Elsevier Materials Today Proceedings,Volume 42, Part 2, 2021, 693-699
Author Response
Thank you very much for your innovative comments and suggestions. Now, we have revised the paper following all of your comments. Please the attached file of "Response to reviewer comments".

Reviewer 2 Report
I received manuscript from Materials MDPI related to manufacturing process of composites laminates using Hand layup and Vacuum infusion. The paper title is “Comparative Manufacturing of Hybrid Composites with Waste Graphite fillers for UAVs”. The paper have less information related to UAV and more information related to comparison between Hand layup vs Vacuum Infusion. Hence, the comparison, the solely method are well established. Many studies are focus on that method and they even add different filler to increase the properties.
If this present paper can go to further publication after the authors also included manufacturing process of UAV structures using 2 different methods and the compared. How UAV body affected by the different manufacturing process? And how the void also take place in different failure mode. The graphite filler also need to be further investigation since many study relies that by adding filler the mechanical properties can increased in optimal point before it goes to decreased.
Another issue related to the manuscript is about replacement existing material using Aluminium with CFRP. Authors should show the part of UAV that using aluminium then replaced by CFRP. The model, complexity and how the process take place also need to be considered.
Other issues that found in the entire manuscript are listed below:
1. In the abstract, the author stated that composites are used to improve the performance of the UAV by reduced it weight. This is a true and many researchers have done it. However, when the authors stated from UAV to manufacturing process, it have no bridging point. That directly jumped to another issue. Even without UAV, the author can state about automotives and or structural building that used CFRP. Why do you used UAV as comparison without doing manufacturing of the UAV parts?
2. In the last paragraph of introduction, the authors take placed the issue of waste toner that used in manufacturing process. If this important, why don’t state that it used to improve the mechanical properties of the laminates?
3. In the materials section, the authors state the materials filler size without any picture to proof the statement. It should be inserted.
4. Preparation specimen better used the actual laminated then the illustrated laminates.
5. In the ratios lists, how the authors mixed the filler with the epoxy hardener?
6. How the authors know that the mixing can make homogenous and the agglomeration is not occurred? No information related to this.
7. The results have less information. Only loading results without any failure picture from the specimen. Are the authors have experiment in actual condition?
8. The failure mode showed in Fig. 10 also need to be detailed by actual picture. Are this finding deals with the experiment?
9. In the conclusion, put the important values from all testing results, characterization, and evaluation. State to the point and direct.

Author Response
Thank you very much for your innovative comments and suggestions. Now, we have revised the paper following all of your comments. Please see the attached file of "Response to reviewer comments".

Reviewer 3 Report
The manuscript entitled "Comparative Manufacturing of Hybrid Composites with Waste Graphite fillers for UAVs" was presented for review.
The following minor revisions are required:
1. It is recommended to show some microscopy images of the raw material and the structure of the composite. Especially fracture surfaces are relevant since you declare the mechanical properties improvement.
2. for Introduction some recent papers about composites can be added:
https://doi.org/10.3390/ma14133530
https://doi.org/10.1016/j.matpr.2022.08.206
https://doi.org/10.1016/j.compstruct.2021.114698
3. lines 154-155 - what was the environment of the curing, please specify.
4. In section 2 more information is needed about the three-point bending test: equipment, according to which standard the test has been performed, etc.
Author Response

(The authors gave the same response as above.)

Round 2
Reviewer 2 Report
We added suggestion in the attachment
